# Vertical and Horizontal Transmission of Pospiviroids

**DOI:** 10.3390/v10120706

**Published:** 2018-12-12

**Authors:** Yosuke Matsushita, Hironobu Yanagisawa, Teruo Sano

**Affiliations:** 1Institute of Vegetable and Floriculture Science, National Agriculture and Food Research Organization, Tsukuba, Ibaraki 305-0852, Japan; 2Central Region Agricultural Research Center, National Agriculture and Food Research Organization, Tsukuba, Ibaraki 305-8666, Japan; yana1208@affrc.go.jp; 3Faculty of Agriculture and Life Science, Hirosaki University, Hirosaki, Aomori 036-8561, Japan; sano@hirosaki-u.ac.jp

**Keywords:** embryo, ovary, ovule, pollen, potato spindle tuber viroid, seed transmission, stigma, style, tomato planta macho viroid

## Abstract

Viroids are highly structured, single-stranded, non-protein-coding circular RNA pathogens. Some viroids are vertically transmitted through both viroid-infected ovule and pollen. For example, potato spindle tuber viroid, a species that belongs to *Pospiviroidae* family, is delivered to the embryo through the ovule or pollen during the development of reproductive tissues before embryogenesis. In addition, some of *Pospiviroidae* are also horizontally transmitted by pollen. Tomato planta macho viroid in pollen infects to the ovary from pollen tube during pollen tube elongation and eventually causes systemic infection, resulting in the establishment of horizontal transmission. Furthermore, fertilization is not required to accomplish the horizontal transmission. In this review, we will overview the recent research progress in vertical and horizontal transmission of viroids, mainly by focusing on histopathological studies, and also discuss the impact of seed transmission on viroid dissemination and seed health.

## 1. Introduction

Viroids are the smallest, self-replicating, non-coding RNAs capable of inducing multiple disease symptoms in susceptible host plants, including potato, tomato, cucumber, hop, coconut, grapevine, fruit trees (apple, avocado, citrus, peach, pear and plum), and some ornamental plants (chrysanthemum and coleus) [1,2]. Viroids are single-stranded RNAs, which range from 250–475 nucleotides in length, and they exist as circular structures with a high degree of self-complementarity either to promote compact folding or to perform their function(s). The short viroid genomes contain all necessary genetic information that enables intracellular trafficking, localization, replication, and pathogenicity of viroids [3]. Worldwide, approximately 30 viroids have been identified and classified into two families, *Pospiviroidae* and *Avsunviroidae* [3]. In *Pospiviroidae* viroids, the secondary RNA structure is either quasi-double-stranded or rod-like, whereas *Avsunviroidae* viroid RNA assumes highly branched secondary structures. Members of the *Pospiviroidae* family, the type species for which is potato spindle tuber viroid (PSTVd), have highly conserved regions in their rod-shaped secondary structure; they replicate in the nuclei of infected cells and lack ribozyme activity [4,5]. *Avsunviroidae* have highly branched structures with self-cleaving ribozymes and replicate in the host chloroplast.

Viroids mainly spread mechanical means: Through infected plant sap, grafting of contaminated scions, plant materials propagated vegetatively from the infected plant and dispersal of infected seeds and pollen. Transmission routes via seeds and pollen are the major ways of infection propagation from parental plants to their progeny and distribution among individuals. The term “seed transmission” means the passage of the pathogen through seeds to seedlings and plants in the next generation. Hence, it is also known as vertical transmission (Figure 1) [6,7,8,9]. Vertical transmission plays an important role in the spread and survival of viroids, as well as viruses. Vertical transmission of viroids has long been known [2,10,11], but much attention has not been paid to this phenomenon until the end of 20th century. Importance of seed transmission of viroids, especially in pospiviroid species, has been highlighted in the early 21st century, when the global distribution of pospiviroid-infected seeds and seed materials became apparent in vegetables and ornamental plants, which increased the risk of diffusion of pospiviroid-infected seeds materials in international trade [12]. The phenomenon of a horizontal transmission closely connected to a vertical transmission, occurs when the pathogen is transmitted to other individuals via pollination with infected pollen (Figure 1) [13,14]. Many plant viruses and viroids are known to spread horizontally or even vertically through pollen in experimental conditions, but for the viroids at least, the effects of these types of transmission on actual plant cultivation have not yet been clearly understood. Recent experimental results suggested that horizontal transmission not only causes spreading of the viroid to the surrounding host of the same species, but also provides a chance to propagate in other host species. Here, we review vertical and horizontal modes of transmission of viroids through pollen, focusing mainly on the recent progress in the histopathological analysis of pospiviroids. For more general aspects of viroid seed and pollen infection, including their economic importance, please refer to the review by Hammond [2].

## 2. Factors Affecting Vertical Transmission of Viroids

Various rates of vertical (or seed) transmission of pospiviroids have been published previously (Table 1). Seed transmission rate varies depending on the species and cultivar identity of the host plant, species, and strain of the viroid, as well as on infection stages and environmental conditions. One of the important factors affecting the transmission rate is host species: For example, the rate of seed transmission of PSTVd was found to be 0.3% in *Capsicum annuum* var. grossum, 0.5% in *C. annuum* var. *angulosum*, and 1.2% in *Glebionis coronaria*. In contrast, seed transmission was not observed in *S. melongena*, *C. annuum*, or *Tagetes patula* [15]. Furthermore, the rate of vertical transmission of PSTVd was found to range from 0 to 90.2% in tomato [15]. Whereas seed transmission rate in tomato cultivar S-4 was about 90%, its value was below 10% in other cultivars. Seed transmission of coleus blumei viroid 1 (CbVd 1) in 14 commercial cultivars of Coleus ranged from 0 to 100% [16]. These reports suggest that vertical transmission of viroids is different depending on host cultivars. Tomato chlorotic dwarf viroid (TCDVd) was reported to be seed-transmissible in the tomato cultivar “Sheyenne” [17], however, no seed transmission was detected in the tomato cultivar “Rutgers” [18]. In the case of CbVd-1, a point-mutation changed the seed transmission of CbVd-1 dramatically [19]. Thus, the differences of viroid strains influence the efficiency of vertical transmission.

Earlier studies in viruses revealed that the time elapsed since infection has a large effect on the seed transmission rate. In particular, the rate of seed transmission in the early stages of virus infection, i.e., late in the growing season, normally results in low or no seed transmission [14,20]. This is also the case in viroids. In PSTVd-infected *Nicotiana benthamiana*, PSTVd did not invade floral and vegetative meristems in the initial stage of infection, but did infect them at the later infection stage [21]. Petunia (*Petunia* × *hybrida*) is known to be highly susceptible, but symptomless, to some pospiviroids infection and is prone to be infected via pollen and seed transmission [13,20]. PSTVd, tomato planta macho viroid (TPMVd), and pepper chat fruit viroid (PCFVd) are efficiently transmitted vertically at a high rate (more than 50%) by pollination with infected pollen (i.e., by “vertical pollen transmission”) in petunia [22,23]. Analysis of seed transmission rate of PSTVd and TPMVd in petunia by pollination with infected pollen grains collected at different time points from 4 to 8 months post-infection revealed that vertical transmission rates increased in both viroids proportionally to the time elapsed since infection [24]. Considering that viroid infection of pollen grains requires the invasion of viroids in the floral meristems before gametogenesis (see below), sufficiently long infection periods are needed for viroids to reach high titers in the infected tissues for invasion into floral meristems.

## 3. Trafficking of Viroids in Floral Organs and Pollen Tubes after Pollination with Infected Pollen

Viral seed transmission depends on whether viruses are transmitted by the infection of the embryo, endosperms, or seed coat [6]. Most seed-transmitted viruses survive in embryos, which leads to systemic infection. Potato true seeds obtained from PSTVd-infected potato plants were infected with PSTVd [30]. In situ hybridization showed that PSTVd was present in the embryo (cotyledon and radical) and endosperm of some tomato cultivars in which PSTVds were highly transmitted by seeds [15]. Whereas some seed-transmitted viruses become inactive in seeds within short periods [7], PSTVds have been recorded to survive for 21 years in true potato seeds [31]. Such long-term survival of viroids in seeds may result from embryonic infection, whereas the infection of seed coat likely results only in low transmission rate. In other words, seed transmission is high or low depending on whether viroids invade the embryo or seed endosperm. Therefore, viroid invasion of the embryo plays a key role in high seed transmission of viroids.

Several pospiviroids can be vertically transmitted by pollen as described above. PSTVd was detected in pollen grains from a PSTVd-infected potato by using return-polyacrylamide gel electrophoresis [30]. PSTVd and TPMVd were detected by RT-PCR in pollen grains obtained from infected petunia plants, and pollination by these pollen grains gave rise to seed transmission rates over 80% and 90%, respectively [32]. Further observation by in situ hybridization showed that TPMVd was present in both the generative and vegetative nuclei of the infected mature pollen grains [33]. At the pollen germination stage, TPMVd was present in the migrating generative nucleus and vegetative nucleus inside the germinating pollen tube on the top of the stigma of the pollinated plants. During pollen tube elongation, TPMVd was present in the vegetative nucleus and also in the two sperm cells in the pollen tube that were generated by the division of the generative nucleus in the style transmitting tract. Pollen transmission of viroids is attributed to the infection of the embryo sac by viroids through infected pollen [22]. Therefore, viroid infection of sperm nuclei is responsible for vertical pollen transmission of viroids. Furthermore, observation by in situ hybridization revealed that PSTVd is also present in the vegetative and generative nucleus of infected pollen grains (Figure 2) [33], suggesting that pospiviroids, due to vertical transmission through pollen, generally invade the generative nucleus. The sporangenous tissue, which gives rise to pollen mother cells, is connected with the anther wall by the tapetum, a layer that provides nutrition to pollen for development [34]. Plasmodesmata between the tapetum and pollen mother cells disappear during meiosis [35]. Since viroid intercellular trafficking occurs through the plasmodesmata, the channels that transverse the cell walls of adjacent cells and do not involve the plasma membrane [36], thus, the presence of the viroid in mature pollen grains suggests that viroids invade pollen mother cells before plasmodesmata disappear. Thus, the presence of the viroid in mature pollen grains suggests that viroids invade pollen mother cells before plasmodesmata disappear. This phenomenon awaits further detailed analysis.

## 4. Mechanism of Seed Transmission

Many seed-transmitted viruses are present in the embryo, and the rate of seed transmission is related closely to embryo infection [6]. Seed transmission of numerous viruses is attributed to viral infection of the embryo directly during embryogenesis or indirectly, before embryogenesis, through the infection of the reproductive tissues (ovule, megaspore mother cell, and pollen mother cell) [13,36]. Indirect virus infection of embryos has been reported based on the observation of virus presence in the megaspore mother cell and egg or in pollen mother cells and pollen [37,38,39,40]. Because the embryo is separated physically from maternal tissues by a callose layer that prevents virus movement through plasmodesmata between cells, the virus must invade the embryo indirectly through reproductive tissues before plasmodesmata disappear and callose barriers develop [14]. These findings indicate that the success of embryo invasion by the virus depends on the developmental stage of the reproductive organs.

PSTVd was detected in all floral parts, such as the sepal, petal, stamen, and pistil of tomato plants [41]. In floral organs of petunia, PSTVd was present in the reproductive tissues of infected plants before embryogenesis [22]. At the floral shoot stage, PSTVd was present in all tissues except for several layers of cells that resemble shoot apical meristems (Figure 3). At the next stage, the viroid was detected in the carpel, petal primordia, stamen primordia, and sepal, but not in ovary primordia. Although PSTVd was absent from the ovule primordium at this stage, it was observed in early developing ovules at the subsequent stage. Finally, PSTVd was present in the ovary wall, placenta, and ovules of PSTVd-infected petunia plants at the flower opening stage (Figure 4). Similarly, the time course in situ hybridization analysis of PSTVd distribution in ovules during different developmental stages leading to seed formation has been performed in ovary parts using PSTVd-infected petunia pollinated with uninfected petunia pollens. This revealed that PSTVd was already distributed in the integuments and parenchyma following the placenta infection at the early stage of embryogenesis and then, at the next stage, PSTVd was observed in the developing endosperm and embryo, and eventually, in the matured seed. Thus, PSTVd strong signals were detected only in the embryo tissue.

The process of flower development is accompanied by drastic changes in cell-to-cell connections between reproductive organs during microsporogenesis and microgametogenesis [42]. For example, the female archesporium and megaspore mother cell have plasmodesmatal connections with the nuclear cells. The functional megaspore and two- and four-nucleate embryo sacs have plasmodesmatal connections with the nucleus. However, the organized embryo sac wall completely lacks plasmodesmatal connections with the surrounding tissue at this time. In the organized embryo sac, the egg, synergids, and central cell are only partially surrounded by cell walls and are in contact only through the plasma membrane [42]. Furthermore, the embryo is separated physically from the mother plant by the callose layer [7,14] and therefore, because viroid intercellular trafficking occurs through the plasmodesmata rather than through plasma membrane-mediated transport [35], viroids cannot directly infect the embryo. Therefore, PSTVd must move into the egg cell in the embryo sac before the plasmodesmata between the embryo sac and placenta tissue have disappeared in the mature ovule stage. Nonetheless, in chili pepper and eggplant, seed transmission of PSTVd was low or absent [15]. Experimental observations showed that PSTVd was present in the placenta and ovary wall, but not in the ovules (Figure 5). Thereby, it was assumed that PSTVd could not invade the egg cells, which resulted in a low rate or no seed transmission.

## 5. Mechanism of Horizontal Transmission

PSTVd, TPMVd, and chrysanthemum stunt viroid (CSVd) are transmitted horizontally by infected pollen (Table 2) [27,30,32]. Tomato plants pollinated with pollen grains infected with CSVd and PSTVd were systemically infected by these viroids [27].

Although the rate of horizontal transmission was very high (more than 80%) in petunia plants when they were pollinated with TPMVd-infected petunia pollen grains, horizontal transmission was not observed at all in petunia pollinated with PSTVd-infected pollen [32]. Comparative histopathological analysis of the distribution of TPMVd and PSTVd in the carpels of petunia plants revealed that when petunia stigmata were pollinated with TPMVd-infected pollen grains, viroids mobilized from pollen tubes to the ovary as pollen tubes elongated to the lower part of style, eventually causing systemic infection of the pollinated plant and horizontal transmission of the viroid [32]. Tissue-printing hybridization and RT-qPCR analysis revealed that TPMVd invaded the ovary via the style through elongating pollen tubes germinated from TPMVd-infected pollen and subsequently spread to the placenta, suggesting that viroid mobilization from the pollen tube to the style and the ovary results in horizontal transmission. In contrast, PSTVd was not detected in the lower part of the style and the ovary in the same conditions. Thus, it failed to infect the pollinated plants systemically, indicating that no horizontal transmission took place. Therefore, the process by which viroid invades from a lower part of the style to the ovary seems to be critical for the establishment of horizontal transmission (Figure 1). Recently, it has been experimentally shown in two plant viruses that the virus genomic RNAs leaked out into a culture medium from the growing pollen tubes [43]. Further analysis is needed whether viroid RNAs also mobilize from the elongating pollen tubes to style tissues.

Furthermore, when TPMVd-infected petunia pollen grains were mixed with healthy tomato pollen to pollinate the stigma of tomato plants, TPMVd was first detected in fruit flesh and subsequently spread systemically in the pollinated plants [32]. In the interspecific-cross of solanaceous plants, germinated pollen tubes can extend into the style and subsequently reach the ovary, however, elongation stops before they enter the ovule [44]. It has been suggested that even if pollen from heterologous plant species cannot fertilize the ovules, if the viroid-infected pollen tubes reach the ovary tissue, viroid infection of the pollinated plants can still occur. These results shed light on a possible mechanism of horizontal transmission by which viroids change the hosts across species barriers in the wild, as well as during farming practices, such as breeding.

## 6. Viroid Nucleotide Sequences Affecting the Efficiency of Horizontal and Vertical Transmission

As viroids are small, self-replicating, non-protein-coding RNAs, single mutations can induce marked changes in their replication, trafficking, pathogenicity, and host range [5,45,46]. This implies that certain genome sequences and/or structures can change the capacity for horizontal and vertical transmission of viroids in the plant invaded mechanically or through pollen. The relationship between the rates of horizontal and vertical transmission on the one hand and viroid sequence and structure on the other hand was examined using several viroid-host combinations. During horizontal transmission through pollen, for example, PSTVd and TPMVd share relatively high (~76%) overall nucleotide sequence homology, however, only TPMVd was highly (~72%) transmitted horizontally in petunia through pollen [32]. In contrast, during vertical (or seed) transmission through pollen, PSTVd and TCDVd share high (ca. 85–90%) overall nucleotide sequence identity, however, only PSTVd invades the ovule and is transmitted through seeds in tomato [18]. In CbVd 1 of the genus *Coleviroid*, a point-mutation at position 25 in loop five from A to UU switched the potential to transmit vertically through coleus seeds [19].

A detailed analysis focusing on the domains of pospiviroid affecting the capacity for horizontal and vertical transmission through pollen has been reported recently using the isolates of TPMVd and PSTVd with high and low horizontal and vertical transmission capacity, respectively, in petunia plants. The pospiviroid genome consists of five domains (terminal left (TL), pathogenicity (P), central, variable (V), and terminal right) [47]. Among the chimeras created by domain-swapping between TPMVd and PSTVd, a chimera with TPMVd-derived TL and P domains was most efficiently horizontally transmitted from infected pollen grains to the style and the ovary by pollination. The chimera with the TL domain of TPMVd showed the second highest horizontal transmission rate. Moreover, these two chimeras harboring the TL domain of TPMVd also had a capacity to transmit vertically at the highest rate by pollination with infected pollen [24]. This result implied that element(s) influencing the capacity of horizontal and vertical transmission through pollination of infected-pollen are mapped to TL and P domains.

## 7. Further Prospects

Seed transmission of viruses depends on the virus strain, host plant species, distribution in seed parts, host environment, and other factors [6,7]. Similarly, the rate of seed transmission of viroids is also dependent on the viroid strain, host plant species, and distribution of viroids in seed parts. As described above in Section 4 of this review, infection of the ovule in the floral organ is important for the establishment of viroid seed transmission [22]. Therefore, preventing viroid invasion of the ovule would block seed transmission. PSTVd has been detected in the ovules of infected tomato and petunia plants but not in the ovules of eggplant and pepper plants [15]. In addition, whereas PSTVd was present in the ovule of infected tomato plants, TCDVd was not [18]. These observations suggest that there are mechanism(s) controlling viroid trafficking from the placenta to the ovule before cytoplasmic connections between them are shut off, which restricts viroid invasion into megaspore mother cell. It is necessary to further analyze the process of viroid invasion into the floral organs to determine whether these differences are (1) controlled by host factors, such as defense mechanism restricting viroid invasion or by transcription factors supporting viroid replication, or (2) depend on the sequence(s) or motif(s) in the viroid molecule regulating replication, accumulation, and trafficking in host plant.

An observation of the disappearance of viroids in pollen tubes elongating in stigma suggests the existence of complex interactions between viroids, pollen (paternal organ), and stigma or pistil (maternal organ). When PSTVd-infected pollen grains pollinate stigmata of uninfected plants and subsequently infected pollen tubes elongate in the pistil, the concentration of PSTVd in the pollen tube gradually decreases during elongation and eventually becomes undetectable even by RT-PCR [32]. In contrast, when pollen grains collected from PSTVd-infected petunia plants were germinated in a liquid medium, viroids in the pollen tubes never disappeared after germination. These results suggest that disappearance or degradation of viroids in elongating pollen tubes is a phenomenon occurring only when pollen tubes extend through the stigma. Viroid RNA molecules are targets of RNA silencing in plant cells: They become a substrate for dicer-like enzymes and are cleaved into small 21–24-nt pieces called short interfering RNAs (siRNA) or viroid-specific small RNAs [5,48]. RNA silencing in viroid-infected pollen grains has been analyzed in hop stunt viroid (HSVd)-infected cucumber [49]. The results indicated that RNA silencing targeting HSVd was actually induced in pollen grains and HSVd-specific small RNAs of the size 21–24-nt were produced as in the vegetative tissues. HSVd infection in pollen grains also caused changes in endogenous small RNAs and induced hypomethylation of rRNA genes and transposable elements in pollen grains, which can change in the transcriptional status in host reproductive tissues. Thus, it is thought that viroid RNAs are degraded and disappear, due to RNA silencing or some other unknown mechanism during pollen tube elongation in the stigma. Therefore, elucidation of the mechanism of viroid disappearance in elongating pollen tubes after pollination during horizontal and vertical pollen transmission will provide important insights about the interaction of viroids and hosts in floral organs. Analysis of the biogenesis of viroid-specific small RNAs in the pollen tubes in floral organs will also provide useful information. However, an important question still remains why only PSTVd but not TPMVd disappears in the pollen tubes during the elongation through the style. Considering that TPMVd was consistently present in the style and ovary, the style may contain a sequence-specific or dose-dependent RNA degradation system whereby PSTVd, but not TPMVd, is degraded.

Some viroids are horizontally transmitted by pollen. However, it remains unclear whether horizontal transmission of viroids indeed occurs in the natural environment or on cultivation agriculture sites. Usually, bumblebees are used to promote pollination in cultivation facilities. Horizontal transmission of viroids through infected pollen carried by bumblebees might also occur naturally, given that both tobacco mosaic virus and pepino mosaic virus are known to be transmitted horizontally in this way [50,51]. In addition, TPMVd was experimentally shown to be transmitted horizontally through pollen between the same and between different plant species. Because bumblebees are important pollinators of multiple plant species, including crops and wildflowers, it should be noted that viroids can be transmitted randomly among multiple plant species through viroid-infected pollen not only by wind but also by insect pollinators.

Consequently, viroids can overcome various barriers existing in the tissues of floral organs to successfully achieve vertical or horizontal transmission. During transmission, viroids are physically and chemically attacked by host defense mechanisms, such as callose deposition, RNA silencing, natural immunity, and others. The interaction between viroids and hosts has been mainly analyzed in vegetative tissues or organs, such as the leaf or stem, in which disease symptoms, e.g., leaf curling, epinasty, and stunting, are most pronounced. In addition to this, it is also important to further analyze the interaction between viroids and hosts in reproductive organs during their development and clarify how viroids establish vertical and horizontal transmission, in order to prevent global epidemics of viroids transmitted through contaminated seeds and seed materials.

## Figures and Tables

**Figure 1 viruses-10-00706-f001:**
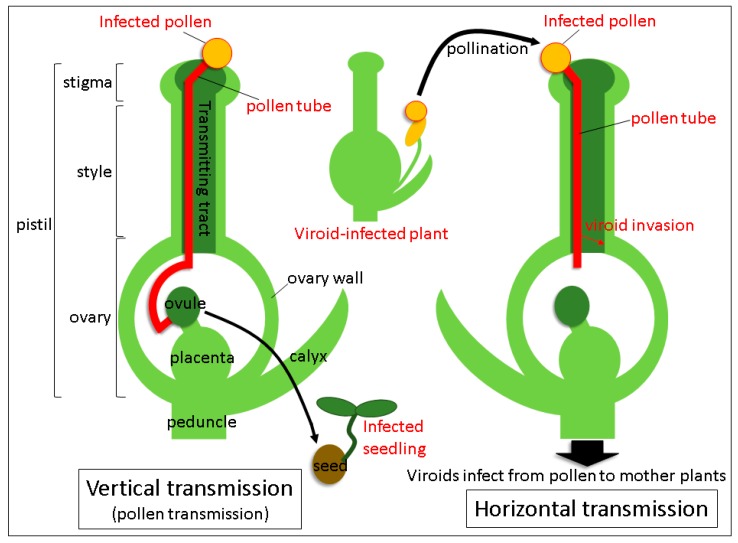
Pathways of tomato planta macho viroid during vertical and horizontal transmission through pollen in infected petunia plants.

**Figure 2 viruses-10-00706-f002:**
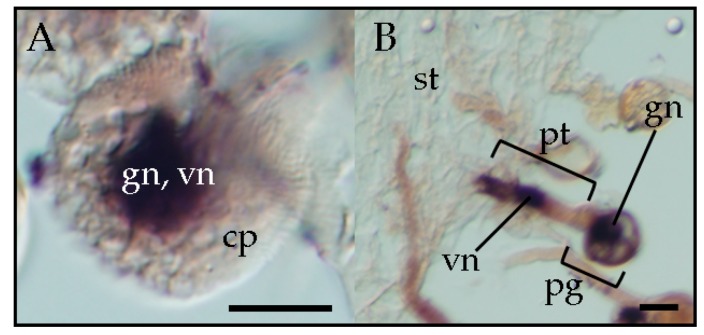
In situ hybridization shows the presence of potato spindle tuber viroid (PSTVd) in the generative nucleus and vegetative nucleus of, respectively, infected mature pollen grains (**A**) and infected germinating pollen grains on healthy stigma (**B**) in PSTVd-infected petunia. cp, cytoplasm; gn, generative nucleus; pg, pollen grain; pt, pollen tube; st, stigma; vn, vegetative nucleus. Scale bars = 50 μm.

**Figure 3 viruses-10-00706-f003:**
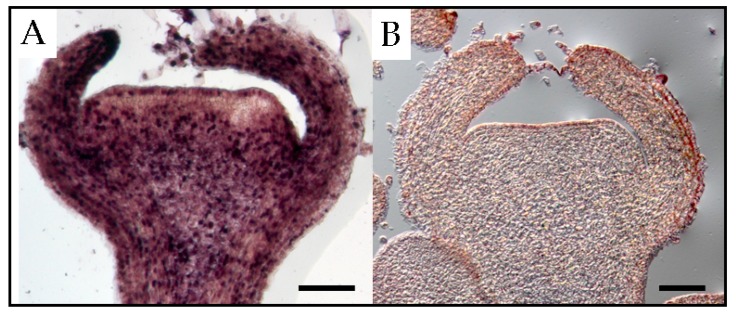
In situ hybridization shows the presence of potato spindle tuber viroid in the floral apical meristem of an infected tomato plant (**A**) and a healthy (**B**) tomato plant. Scale bars = 50 μm.

**Figure 4 viruses-10-00706-f004:**
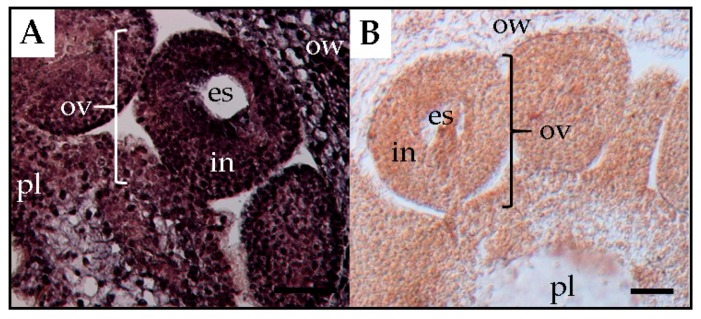
In situ hybridization shows the presence of potato spindle tuber viroid in the placenta, ovule, and ovary wall in flowers of an infected tomato plant (**A**) and a healthy tomato plant (**B**) at the flower opening stage. es, embryo sac; in, integuments; ov, ovule; ow, ovary wall; pl, placenta. Scale bars = 50 μm.

**Figure 5 viruses-10-00706-f005:**
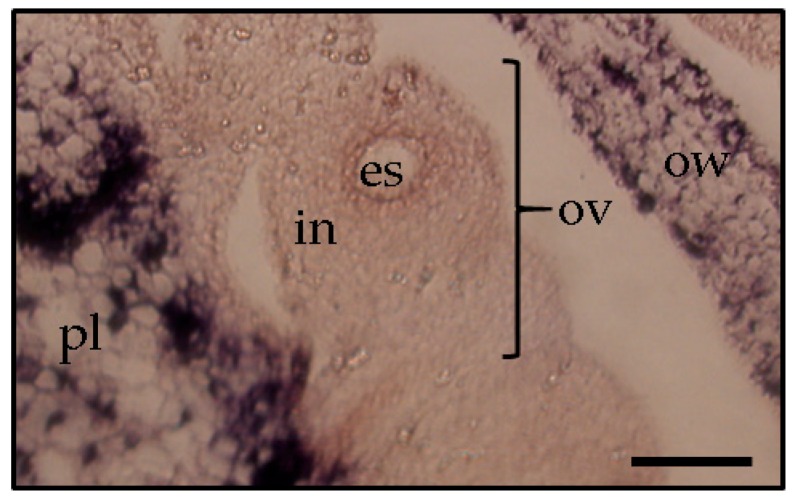
In situ hybridization shows the presence of potato spindle tuber viroid in the ovary wall (ov) and placenta (pl), but not the ovule (ov), which comprise the embryo sac (es) and integument (in), respectively, in a flower of an infected eggplant (*Solanum melongena*) at the flower opening stage. Scale bars = 100 μm.

**Table 1 viruses-10-00706-t001:** Seed- and pollen-transmitted pospiviroids.

Viroid	Host Plant Species	Seed-Transmitted	Pollen-Transmitted	Reference
Potato spindle tuber viroid	Potato	+	+	[11,25,26]
	Tomato	+	+	[10,11,15,27]
	Capsicum annum	+		[15]
	Glebionis coronaria	+		[15]
	Petunia hybrida	+	+	[22,23]
Tomato chlorotic dwarf viroid	Tomato	+		[17]
	Petunia hybrida	+		[15]
Tomato apical stunt viroid	Tomato	+		[15]
Tomato planta macho viroid	Tomato	+		[23]
	Petunia hybrida	+	+	[23]
Pepper chat fruit viroid	Tomato	+		[23]
	Capsicum annum	+		[28]
	Petunia hybrida	+	+	[23]
Columunea latent viroid	Tomato	+		[15]
Chrysanthemum stunt viroid	Tomato	+	+	[27]
	Chrysahtmemum	+	+	[16]
Citrus exocortis viroid	Tomato	+		[29]
	Impatiens	+		[29]
	Verbena	+		[29]

**Table 2 viruses-10-00706-t002:** Horizontal transmission of pospiviroids.

Viroid	Host Plant Species	Reference
Potato spindle tuber viroid	Potato	[30]
	Tomato	[27]
Tomato planta macho viroid	Petunia hybrida	[32]
Chrysanthemum stunt viroid	Tomato	[27]

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
