# Peer review of "Vertical and Horizontal Transmission of Pospiviroids"

_viruses, 2018, doi:10.3390/v10120706_

Round 1

Reviewer 1 Report

The manuscript reviews vertical and horizontal modes of transmission of viroids through pollen. Viroids are important viral pathogens infecting plants and cause important economic losses. This review manuscript well addresses the mechanism of viroid transmission. I have only a few minor comments:

1- In the Introduction, viroid names are italicized whereas they are not in the other sections.

2- Abbreviations for several viroids are not defined consistently before they are used (ex. on line 77: PSTVd, line 81: TCDVd …)

3- The TL and P domains should be defined and explained (Section 6).

Author Response

(To Reviewer 1#)

1- In the Introduction, viroid names are italicized whereas they are not in the other sections.

As) corrected  (L40)

2- Abbreviations for several viroids are not defined consistently before they are used (ex. on line 77: PSTVd, line 81: TCDVd …)

As) corrected (L40, L84)

3- The TL and P domains should be defined and explained (Section 6).

As) corrected. I added to the explain about these words and added its reference[44] (L 266-267).

Reviewer 2 Report

In this manuscript, the authors summarize the latest knowledge about vertical and horizontal transmission of pospiviroids.  The article is interesting and very well written.

Author Response

>In this manuscript, the authors summarize the latest knowledge about vertical and horizontal transmission of pospiviroids.  The article is interesting and very well written

As) Thank you for your comments.

Reviewer 3 Report

In this review article Matsushita et al review the topic of vertical and horizontal transmission of pospiviroids. This review article is a much needed update in the field since there are very few manuscripts regarding this topic. The review is well written and structured. Nevertheless there are a few changes that I think will help to make the manuscript even better:

1) I think that focusing the review to the pospiviroidae family reduces the target audience for this review, although I understand that this is the expertise of the authors. In order to aim for a wider audience, can the authors include also the examples of vertical and horizontal transmission of members of the avsunviroidae family? I think this will increase the interest from other members of the virology field to this review.

2) Regarding the analysis of the RNA silencing activity against viroids in the pollen grain, recently a manuscript analyzing the presence and activity of this pathway in the pollen grain of cucumber plants infected with the hop stunt viroid was published (https://academic.oup.com/jxb/article/67/19/5857/2236510). The authors should include these results to this review, since it actually analyzes the production of viroid-derived sRNAs in the pollen grain, but also shows that HSVd can also accumulate in the pollen grain of infected plants.

Minor changes:

1) In the abstract the examples cited need to be better connected to the text. The use of the header "for example," could help to present the examples shown in a more fluent way.

2) Line 43: Dispersal of infected seeds and pollen is included in transmission through mechanical means. I would not consider these to be mechanical transmission of viroids.

3) Line 45: "transmission through insect vectors rarely occurs". To my knowledge there is only one example of this transmission (TPMVd) so it should be mentioned.

4) Line 83: Can the two explanations of the discrepancies be joined in the same sentence? It gives a more natural flow for the reader.

5) Line 94: "pepper chat fruit viroid" needs to have an acronym too.

6) Lines 131 to 136. Can the authors re-write this part? I think it is more understandable to justify that the presence of viroids in the mature pollen grain needs to start at the pollen mother cell stage by placing the last sentence first. I suggest this change: "The sporangenous tissue, which gives rise to pollen mother cells, is connected with the anther wall by the tapetum, a layer that provides nutrition to pollen for development [31]. Plasmodesmata between the tapetum and pollen mother cells disappear during meiosis [32]. Since viroid intercellular trafficking occurs through the plasmodesmata, the channels that transverse the cell walls of adjacent cells and do not involve the plasma membrane [30], thus, the presence of the viroid in mature pollen grains suggests that viroids invade pollen mother cells before plasmodesmata disappear."

Author Response

(Reviewer 3#)

1) I think that focusing the review to the pospiviroidae family reduces the target audience for this review, although I understand that this is the expertise of the authors. In order to aim for a wider audience, can the authors include also the examples of vertical and horizontal transmission of members of the avsunviroidae family? I think this will increase the interest from other members of the virology field to this review.

As) Thank you for your suggestion. At first, I tried to write contents about transmission of avsunviroidae. However, there are a few reports of avsunviroidae. So, I did not add the contents. As a result, the review mainly shows pospiviroid of vertical and horizontal transmission.

2) Regarding the analysis of the RNA silencing activity against viroids in the pollen grain, recently a manuscript analyzing the presence and activity of this pathway in the pollen grain of cucumber plants infected with the hop stunt viroid was published (https://academic.oup.com/jxb/article/67/19/5857/2236510). The authors should include these results to this review, since it actually analyzes the production of viroid-derived sRNAs in the pollen grain, but also shows that HSVd can also accumulate in the pollen grain of infected plants.

As) I added to the sentence (L303-308) of ‘RNA silencing in viroid-infected pollen grains has been analyzed in Hop stunt viroid (HSVd)-infected cucumber [46]. The results indicated that RNA silencing targeting HSVd was actually induced in pollen grains and HSVd-specific small RNAs of the size 21–24-nt were produced as in the vegetative tissues. HSVd infection in pollen grains also caused changes in endogenous small RNAs and induced hypomethylation of rRNA genes and transposable elements in pollen grains, which can change in the transcriptional status in host reproductive tissues.’

Minor changes:

1) In the abstract the examples cited need to be better connected to the text. The use of the header "for example," could help to present the examples shown in a more fluent way.

As) corrected.(In Abstract)

2) Line 43: Dispersal of infected seeds and pollen is included in transmission through mechanical means. I would not consider these to be mechanical transmission of viroids.

As) corrected the sentence (L44-46)

3) Line 45: "transmission through insect vectors rarely occurs". To my knowledge there is only one example of this transmission (TPMVd) so it should be mentioned.

As) deleted the sentence of insect transmission

4) Line 83: Can the two explanations of the discrepancies be joined in the same sentence? It gives a more natural flow for the reader.

As) deleted the sentence

5) Line 94: "pepper chat fruit viroid" needs to have an acronym too.

As) corrected (L97)

6) Lines 131 to 136. Can the authors re-write this part? I think it is more understandable to justify that the presence of viroids in the mature pollen grain needs to start at the pollen mother cell stage by placing the last sentence first. I suggest this change: "The sporangenous tissue, which gives rise to pollen mother cells, is connected with the anther wall by the tapetum, a layer that provides nutrition to pollen for development [31]. Plasmodesmata between the tapetum and pollen mother cells disappear during meiosis [32]. Since viroid intercellular trafficking occurs through the plasmodesmata, the channels that transverse the cell walls of adjacent cells and do not involve the plasma membrane [30], thus, the presence of the viroid in mature pollen grains suggests that viroids invade pollen mother cells before plasmodesmata disappear."

As) corrected (L 136- 142)
